# Slotted Antenna Array with Enhanced Radiation Characteristics for 5G 28 GHz Communications

**Ahmed A. Ibrahim** [1], **Hijab Zahra** [2], **Osama M. Dardeer** [3], **Niamat Hussain** [4,*], **Syed Muzahir Abbas** [2,*] **and Mahmoud A. Abdelghany** [5,1]

1. Electrical Engineering Department, Faculty of Engineering, Minia University, El-Minia 61519, Egypt
2. School of Engineering, Faculty of Science and Engineering, Macquarie University, Sydney NSW 2109, Australia
3. Microstrip Department, Electronics Research Institute (ERI), El Nozha, Cairo 12622, Egypt
4. Department of Smart Device Engineering, Sejong University, Seoul 05006, Korea
5. Electrical Engineering Department, College of Engineering, Prince Sattam Bin Abdulaziz University, Wadi Addwasir 11991, Saudi Arabia
* Correspondence: niamathussain@sejong.ac.kr (N.H.); syed.abbas@mq.edu.au (S.M.A.)

**Abstract:** This paper presents a $1 \times 4$ linear antenna array working at 28 GHz for 5G communication systems. The proposed array employs four rectangular slotted antenna elements fed by a $1 \times 4$ T-power divider. An artificial magnetic conductor (AMC) layer is placed below the array for increasing the radiation intensity and improving overall array gain. The measured impedance bandwidth of the proposed array with ($|S_{11}| < -10$ dB) is extended from 25.36 to 26.03 GHz (with a bandwidth of 0.67 GHz) and from 26.75 to 28.81 GHz (with a bandwidth of 2.06 GHz). The proposed array design exhibits a measured gain value that varies between 11.8 dBi and 13.1 dBi within the operating bands and reaches 13.1 dBi at 28 GHz. The proposed array achieves a radiation efficiency of 83.05%, and a front-to-back ratio ranging between 15 and 20 dB across the operating frequency band. The array is fabricated and tested with good matching between the simulated and tested outcomes. The improved performance of the array makes it a suitable candidate for 5G new radio (NR) communications.

**Keywords:** slotted antenna array; T-power divider; artificial magnetic conductor (AMC); gain improvement; 5G applications; millimeter wave

## 1. Introduction

The mobile traffic volumes will cause remarkable challenges to the mobile access networks shortly. More than ten times more than nowadays will be the increment of the number of connected devices. The transition from microwave to millimeter-wave (MMW) frequency bands may be achieved in gradual steps to take advantage of the higher capacity, decreasing the system latency, higher bandwidth, as well as a large amount of available spectrum [1–4]. There are many frequency bands assigned by the FCC [3,5]. The 5G NR has many standards around 28 GHz, such as n257 (LMDS), n258 (K-band), and n261 (Ka-band). Antennas operated at these frequency bands with special features are needed. Because of the signal attenuation caused by oxygen molecules' absorption at these frequency bands [6], a high gain antenna should be utilized [7–10].

Several antenna designs with high gain are introduced by the researchers [10–18]. In [10], an array antenna with a dense dielectric patch working at 28 GHz is discussed. The radiation efficiency of the dense dielectric patch antenna is higher than the traditional metallic patch antenna, especially at higher frequencies. A superstrate is adopted, which acts as a lens for increasing the radiation intensity in a certain direction which in turn increases the antenna gain significantly as discussed in [11]. An 8-element Yagi-Uda antenna with again around 12.7 at 28 GHz is investigated in [12]. In [13], a 5-element linear

array with peak gain equal 10 dBi and operating at 28 GHz is studied. An 8-element dipole antenna with a peak gain of around 12.5 dBi is discussed in [14].

The AMC as a reflector can be used to improve the overall antenna gain instead of the perfect electric conductor (PEC) reflector because the AMC is placed at a distance lower than $1/4 \lambda_0$, which decreases the overall size of the antenna. The AMC has an in-phase reflection, which behaves like a perfect magnetic conductor (PMC). Additionally, at the designed frequency band, the AMC can behave like a high impedance surface [19]. The proposed reflection coefficient and the antenna gain can be controlled by controlling the AMC shape, the spacing from the antenna, and the AMC size. Several AMC array cells embedded below the antenna have been discussed in [20–24]. In [20], the AMC-based single patch antenna fed by the coaxial cable, and operating at 30 GHz, and a peak gain of 9 dBi is investigated. The AMC-based single bowtie antenna operating at 33 GHz with a peak gain of 5.5 dBi is discussed in [21]. In [22], a single patch antenna embedded with AMC cells for antenna-in-package application and operating at 30.8 GHz with a peak gain of 7.8 dBi is studied. In addition, the AMC-based antennas for sub-6 GHz and UWB applications are discussed in [23,24].

In this paper, a $1 \times 4$ linear antenna array is proposed for 5G NR networks. The antenna is a rectangular slot with a parasitic element for impedance bandwidth tuning. An AMC layer is placed below the array for gain improvement at the 28 GHz band. The array is fed by a $1 \times 4$ T-power divider. The proposed design offers compact size, increased gain characteristics, and a simple feeding structure with a good trend between the simulated and tested outcomes.

The paper is arranged as follows. Section 2 discusses the configuration of a single antenna, $1 \times 2$ array, and $1 \times 4$ array. The AMC layer is explained in Section 3. The results of the final design are discussed in Section 4. Finally, the conclusion of the paper is written in Section 5.

## 2. Antenna Design

### 2.1. Single Element Antenna

Figure 1 shows the geometry of the printed rectangular slot antenna, which is designed on a Rogers RT/Duroid 4003 substrate ($\varepsilon_r$ = 3.55, h = 0.203 mm, and tan δ = 0.0027). The antenna is composed of a 50-Ω microstrip feeding line, a rectangular slotted ground plane, and a parasitic rectangular element with chamfered corners. The feed line is on the top layer of the substrate, while the slotted ground and the parasitic element are on the bottom layer.

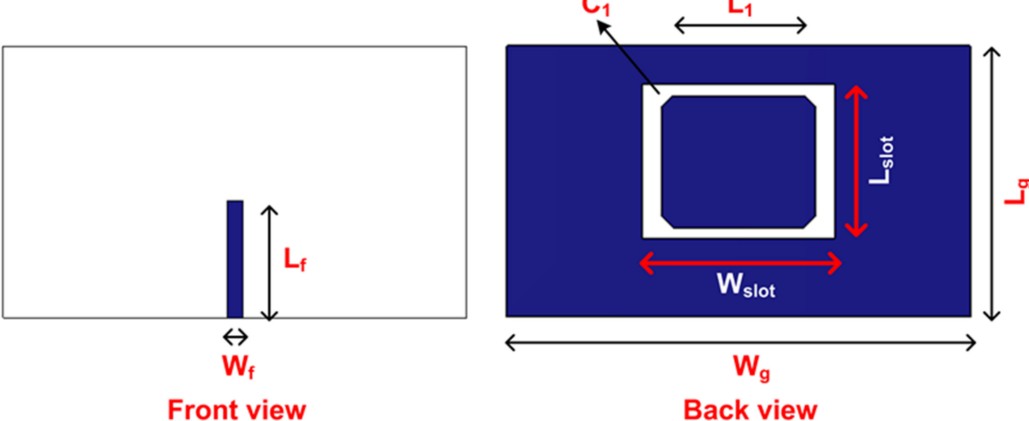

**Figure 1.** The geometry of the proposed rectangular slotted antenna with a parasitic element.

The length of the feeding line is a crucial design parameter for obtaining good matching characteristics, since the field lines are formed at the end of this line towards the slot [25,26]. Any specific length represents inductance and capacitance values which may affect the matching condition. Therefore, the chamfered rectangular parasitic element is added for

enhancing the matching characteristics and tuning a large number of resonating modes. The antenna is optimized to achieve the compact size and can be simply arranged in a linear array for gain enhancement. The total antenna size is $12 \times 7 \times 0.203$ mm$^3$. The design parameters are: Wf = 0.4, Lf = 3, Wg = 12, Lg = 7, Wslot = 5, Lslot = 4, L1 = 3.4, and C1 = 0.3 (all in millimeters).

The proposed antenna has been investigated utilizing the CST Microwave Studio. The reflection coefficient and gain characteristics of the proposed antenna are displayed in Figure 2. The realized frequency band (for $|S_{11}| < -10$ dB) is 26.62–29.52 GHz (with a bandwidth of 2.9 GHz) applied for the new radio band of 5G communications. The proposed antenna provides front and back radiation due to the existence of the slot inside the ground plane. The proposed antenna has gain, directivity, and radiation efficiency of 4.43 dBi, 4.69 dBi, and 94.16%, respectively, at 28 GHz.

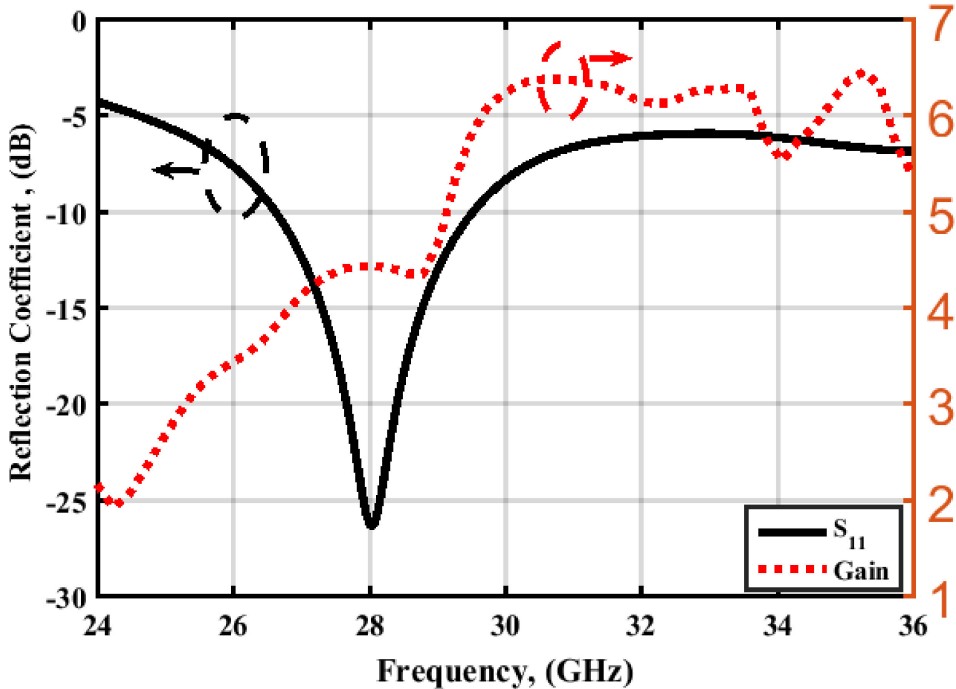

**Figure 2.** Reflection coefficient and gain characteristics of the single slotted antenna.

### 2.2. $1 \times 2$ Sub-Array

To constitute a $1 \times 4$ linear array with improved gain performance depending on the previous antenna, a feeding network with a simple equal phase is utilized. A $1 \times 2$ sub-array is designed in this sub-section which is repeated three times to complete a $1 \times 4$ feeding network. A complete $1 \times 4$ antenna array will be presented in the next sub-section.

To minimize the losses and increase the gain performance, a simple T-power divider is utilized to feed the array elements with matched performance and minimum losses. There is no need for an isolation resistor in this design, as in the case of a Wilkinson power divider. Two 50-Ω lines are connected in parallel to achieve a 25-Ω impedance which requires a λ/4 transformer section with 35.35-Ω impedance to be connected to the 50-Ω feeding line. This is clearly illustrated in Figure 3. The geometry and S-parameters of the T-power divider are shown in Figure 3.

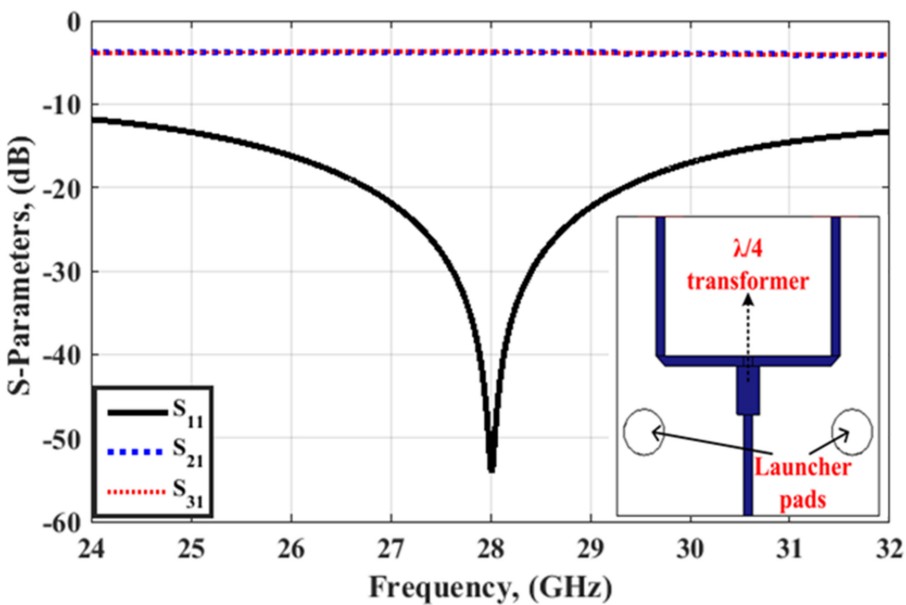

**Figure 3.** Geometry and S-parameters of the T-power divider for array feeding.

A 1 × 2 antenna array is developed depending on the previous antenna. The spacing between the elements is 8 mm, which is equal to 0.746 $\lambda_0$ at a frequency of 28 GHz. The two elements are excited through a T-power divider that is presented earlier, as displayed in Figure 4. Figure 5 shows the reflection coefficient and gain characteristics of the proposed 1 × 2 array. The achieved frequency band (for |S$_{11}$| < −10 dB) is 26.35–30.15 GHz (3.8 GHz bandwidth) for the new radio band of 5G communications. The 1 × 2 array provides gain, directivity, and radiation efficiency values as 7.73 dBi, 8.06 dBi, and 92.68 %, respectively, at 28 GHz.

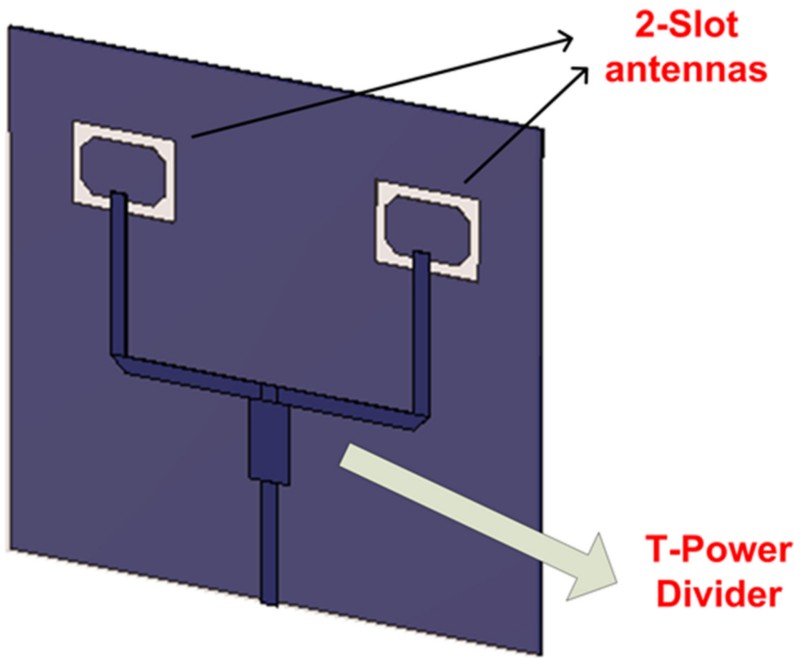

**Figure 4.** The geometry of the 1 × 2 sub-array.

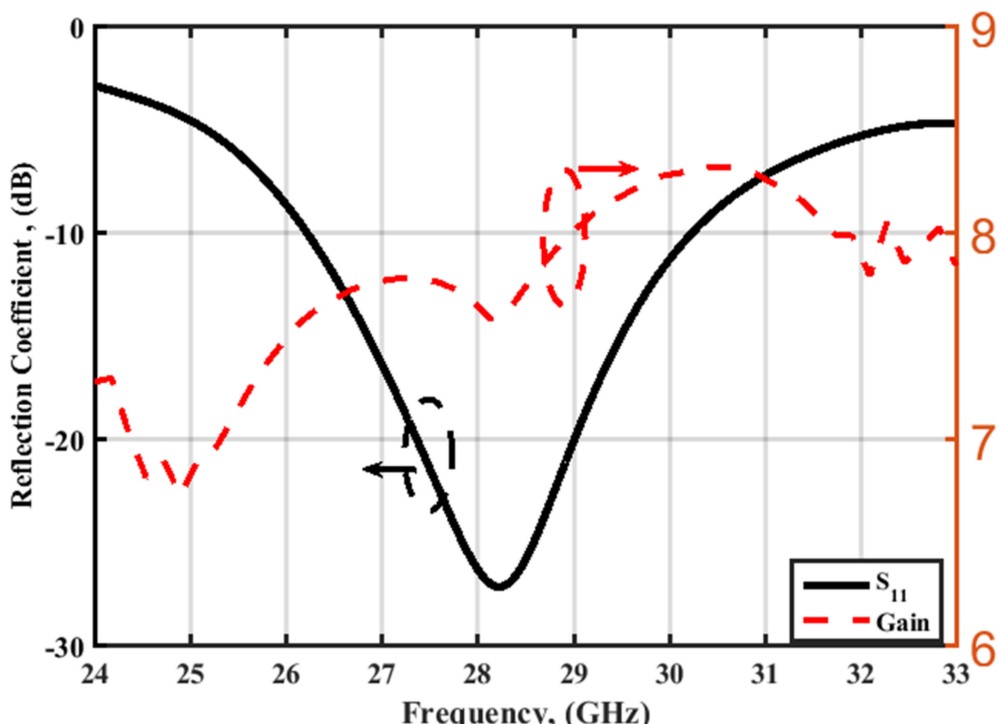

**Figure 5.** Reflection coefficient and gain characteristics of the 1 × 2 sub-array.

### 2.3. 1 × 4 Antenna Array

A 1 × 4 antenna array has been developed by using three units of the previous T-power divider as a 1 × 2 array. The spacing between the elements is 8 mm which equals 0.746 $\lambda_0$ at a frequency of 28 GHz. Different optimization trials have been carried out using the particle swarm optimization (PSO) tool in the CST simulator to achieve good impedance matching and acceptable gain characteristics. The optimization parameters are the $\lambda/4$ sections in the T-dividers, the spacing between the array elements, and the feeding line length before the four slot antennas.

Figure 6 illustrates the geometry of the 1 × 4 array with the launcher connector. The antenna has a 30 × 22.1 mm$^2$ total size. Figure 7 displays the fabricated array photo. The reflection coefficient characteristics of the 1 × 4 antenna array are illustrated in Figure 8. The Rohde and Schwarz ZVA 67 is utilized in the measurement process. The achieved frequency band for the simulated results (for $|S_{11}| < -10$ dB) is 26.3–29.03 GHz (with a bandwidth of 2.73 GHz). Furthermore, the measured results have an achieved frequency band (for $|S_{11}| < -10$ dB) of 25.79−28.7 GHz (with a bandwidth of 2.91GHz) suitable for 5G NR communications. The two results are matched well with a slight shift between them; this is due to the fabrication process which cannot be overcome. Figure 9 illustrates the gain variation versus frequency for the 1 × 4 array. The simulated peak gain of the array agrees well with the measured results. The simulated gain, directivity, and radiation efficiency at 28 GHz achieve 10.7 dBi, 10.96 dBi, and 90.07%, respectively. In addition, the measured gain has 10.3 dBi as shown in Figure 9. The 3D and 2D normalized radiation patterns at 28 GHz are shown in Figure 10. The 1 × 4 array provides a bidirectional pattern in the two planes with small beamwidth which confirms the high directivity achieved by the array elements. A good trend between the two results is accomplished.

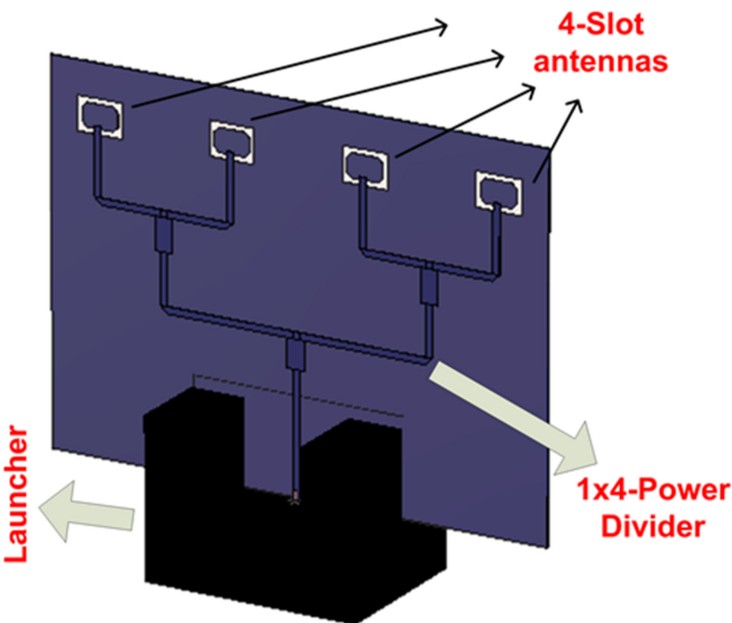

**Figure 6.** The geometry of the 1 × 4 antenna array.

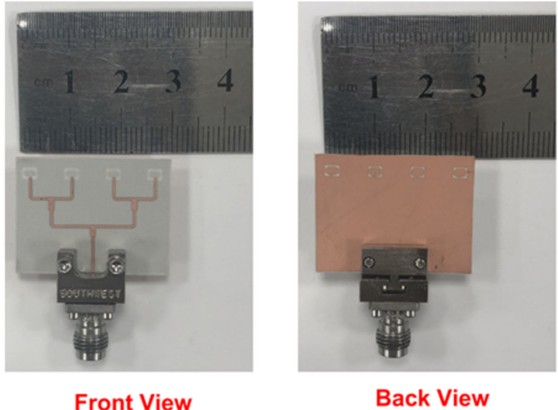

**Figure 7.** The fabricated 1 × 4 antenna array photo.

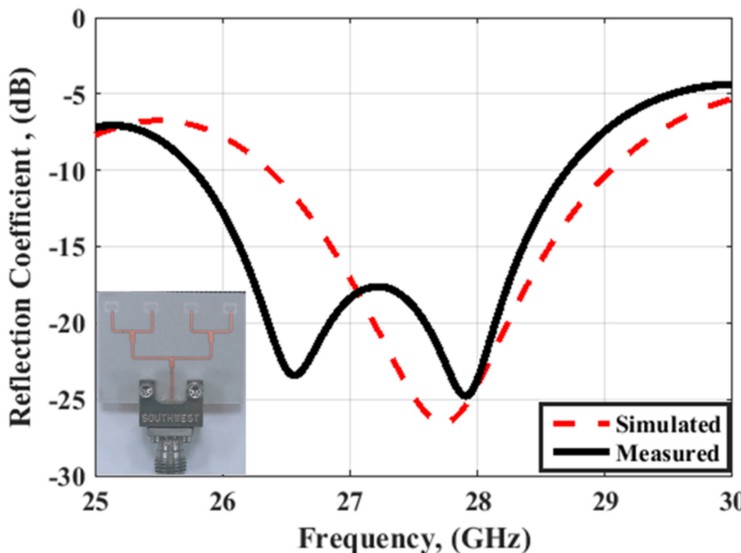

**Figure 8.** Reflection coefficient simulated and measured results of the 1 × 4 antenna array.

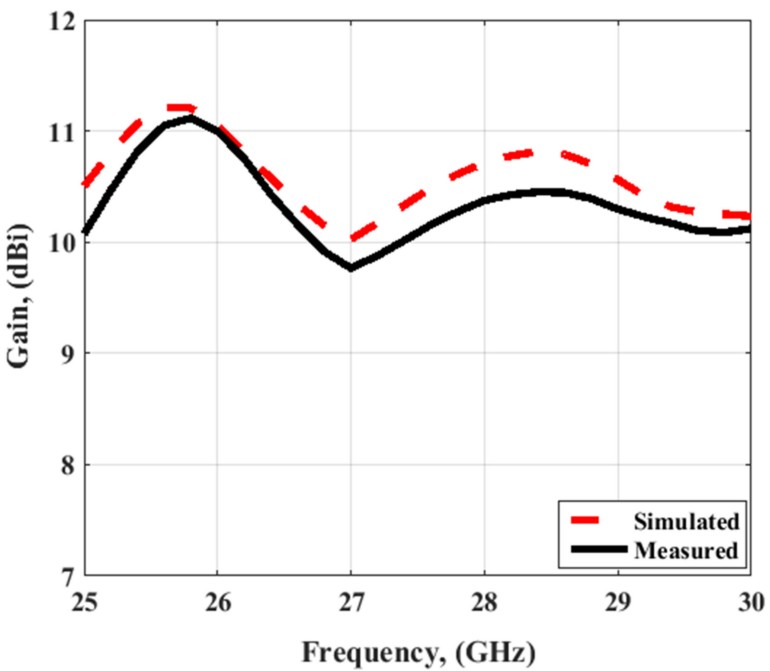

**Figure 9.** Gain variation versus frequency for the 1 × 4 antenna array.

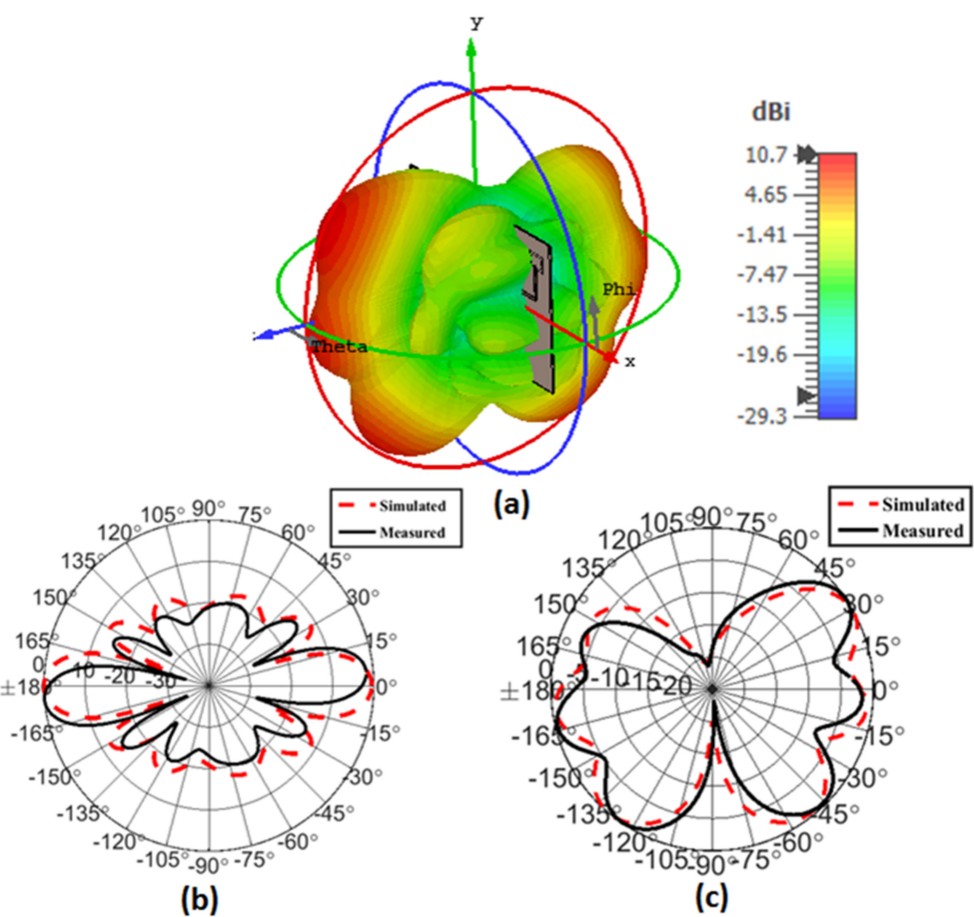

**Figure 10.** Radiation pattern of the 1 × 4 antenna array at 28 GHz; (**a**) 3D, (**b**) normalized pattern @ XZ-plane, and (**c**) normalized pattern @ YZ-plane.

## 3. AMC Layer for Gain Improvement

The AMC layer is used to enhance the gain of the antenna by inserting the AMC layer below the proposed antenna because it introduces an in-phase reflection at a specific frequency band. The configuration of the AMC unit cell with the final dimensions is illustrated in Figure 11a. The suggested AMC cells are printed above the Rogers RT/5880 substrate ($\varepsilon_r$ = 2.2, h = 0.787 mm, and tan δ = 0.0009). Two layers of copper with a thickness of 0.035 mm are introduced. The I-shape is etched on the top of the substrate while a complete ground is added on the back. The I-shaped size is optimized to achieve the desired results around 28 GHz. The optimized cell size of the 6 × 6 AMC array with 39.5 mm × 39.5 mm as shown in Figure 11c is suggested to cover the proposed antenna. The fabricated photo of the 6 × 6 AMC array is shown in Figure 11d. Two PMC walls are utilized in the ±x direction while two PEC walls are used in the ±y direction as a boundary condition to produce the $S_{11}$ reflection phase behavior of the AMC unit cell as shown in Figure 12. The unit cell produces 0° phase at 28 GHz with ±90° at 27.05 GHz and 28.7 GHz, respectively, as illustrated in Figure 12. At 28 GHz, the current distribution of the unit cell is studied as shown in Figure 11b and shows that the current is collected around the outer edges of the etched I-shape.

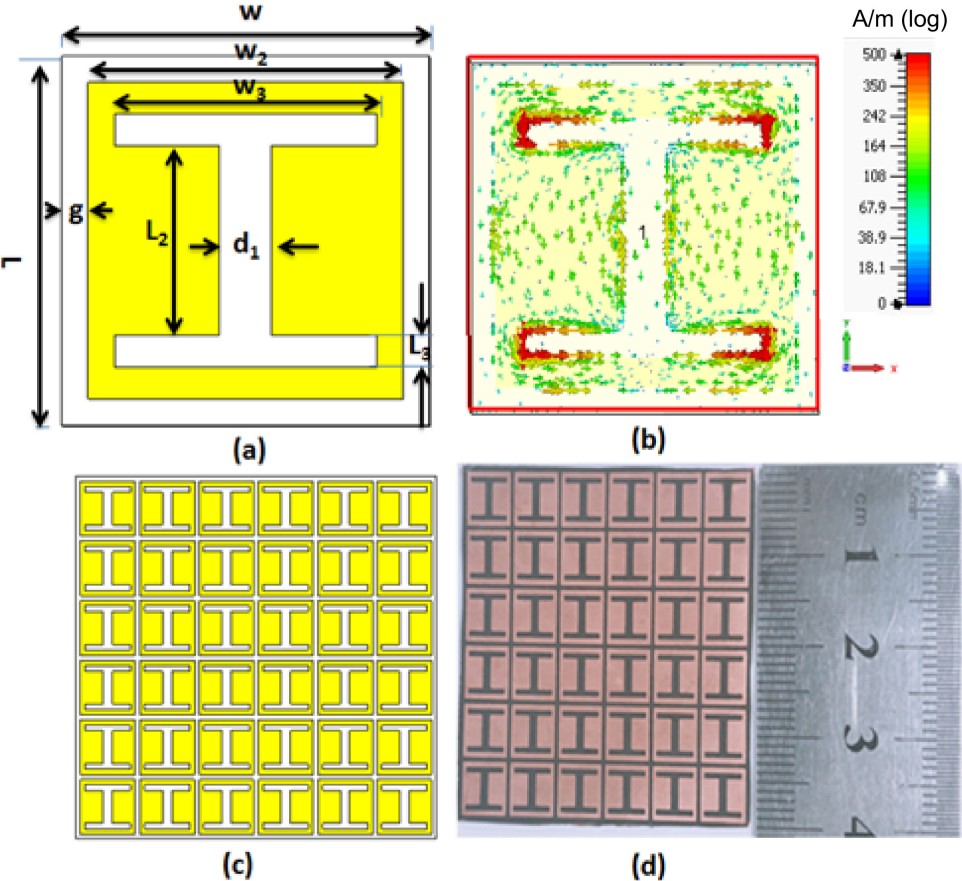

**Figure 11.** Proposed structure of AMC (**a**) 2D configuration of the unit cell with (w = L = 7 mm, w2 = 6 mm, w3 = 5 mm, g = 0.5 mm, L2 = 3.6 mm, d1 = 1 mm, and L3 = 0.6 mm). (**b**) Current distribution @ 28 GHz of the unit cell; (**c**) a 6 × 6 array of AMC unit cells; (**d**) fabricated photo of the proposed AMC.

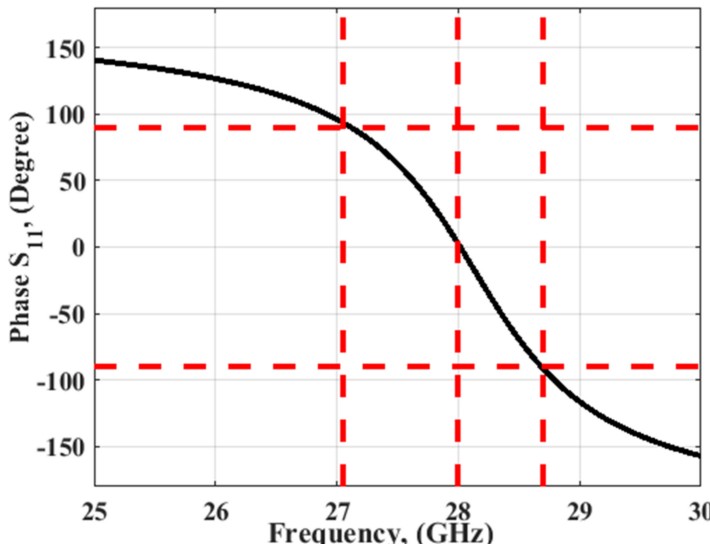

**Figure 12.** The phase of $|S_{11}|$ results of the proposed AMC unit cell.

## 4. Results and Discussion

The proposed antenna array has been developed by placing the AMC layer discussed in Section 3 below the $1 \times 4$ array discussed in Section 2 for gain enhancement and removing the back radiation as much as possible. Different optimization trials have been carried out to reach the best gain characteristics. In this case, the optimization parameters are the separation between the array and the AMC layer and the AMC array size. The separation between the antenna and the AMC is 4.7 mm.

The geometry of the proposed antenna array with AMC is illustrated in Figure 13. A photograph of the fabricated antenna array is shown in Figure 14. The reflection coefficient characteristics of the proposed antenna array are displayed in Figure 15. The achieved frequency band for the simulated results (for $|S_{11}| < -10$ dB) is 24.8−25.94 GHz (1.14 GHz bandwidth) and from 26.59−28.63 GHz (2.04 GHz bandwidth). Furthermore, the measured results achieved a frequency band (for $|S_{11}| < -10$ dB) of 25.36-26.03 GHz (with a bandwidth of 0.67 GHz) and from 26.75-28.81 GHz (with a bandwidth of 2.06 GHz). The two results are matched well with a slight shift between them; this is due to the fabrication process and the AMC alignment.

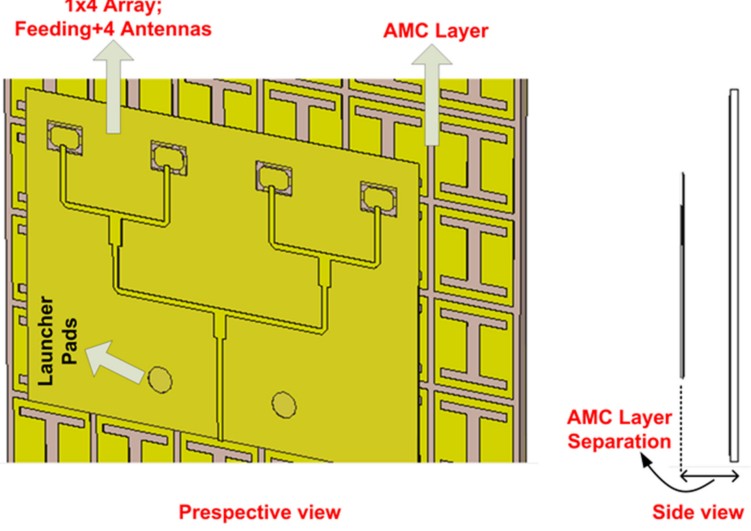

**Figure 13.** Perspective and side views of the embedded antenna with AMC.

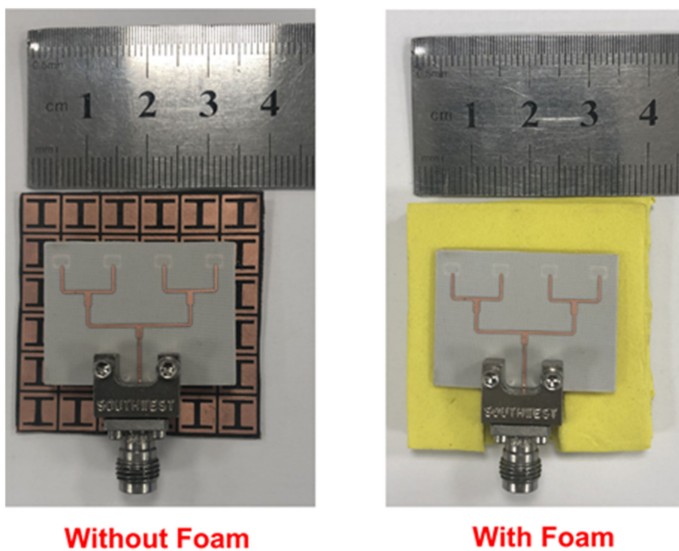

**Figure 14.** The fabricated antenna photo with AMC.

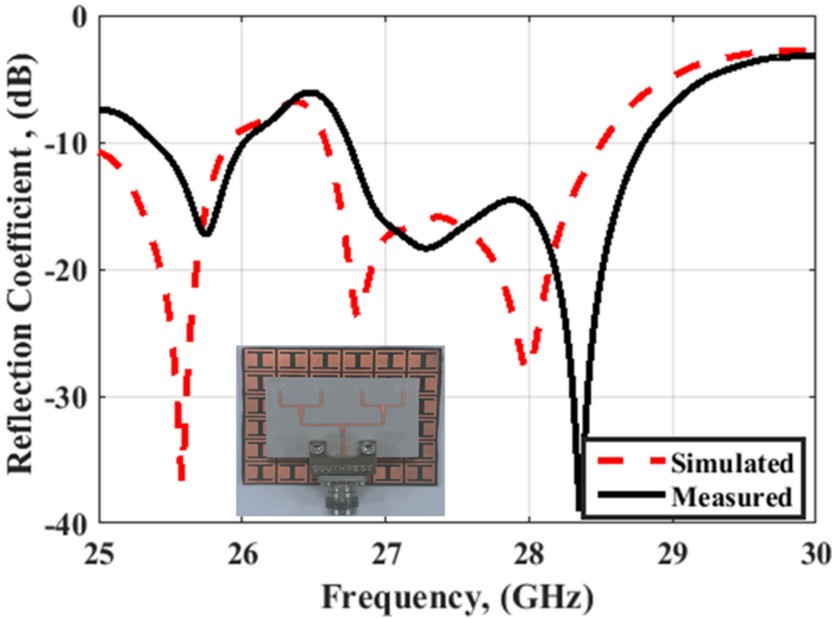

**Figure 15.** Reflection coefficient of the embedded antenna with AMC.

Figure 16 illustrates the gain variation versus frequency for the proposed array.

The simulated gain, directivity, and radiation efficiency at 28 GHz achieve 13.4 dBi, 14.2 dBi, and 83.05%, respectively. The reduction of the radiation efficiency is due to increasing the losses introduced by the AMC array cells. In addition, the measured gain varies between 11.8 dBi and 13.1 dBi within the operating bands and reaches 13.1 dBi at 28 GHz as shown in Figure 16. The normalized radiation patterns at 28 GHz are provided in Figure 17. It can be observed that the proposed array greatly reduces the back radiation and the front-to-back ratio ranges between 15 and 20 dB across the operating frequency band, which confirm the high directivity achieved by the AMC array cells. A good trend between the two results is accomplished.

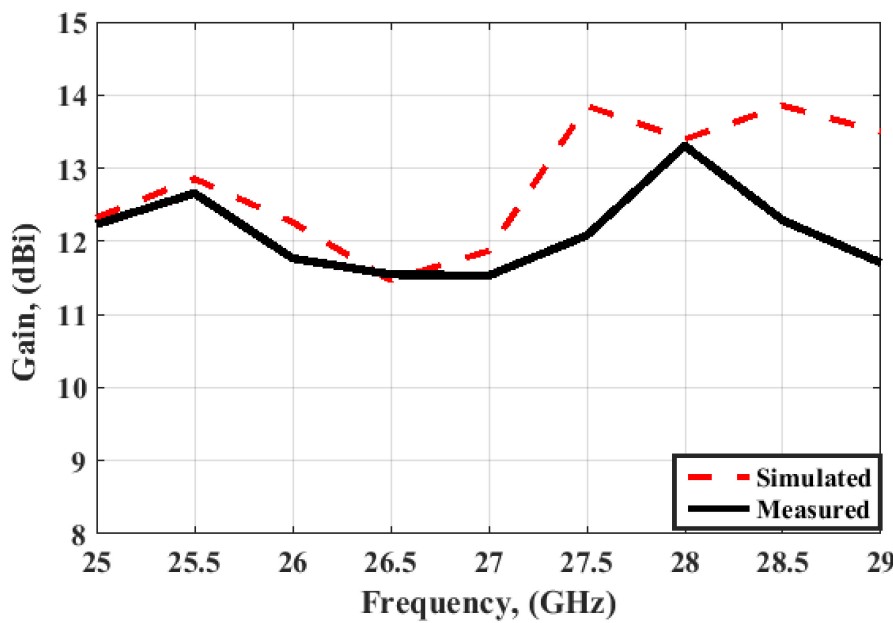

**Figure 16.** Gain variation versus frequency of the antenna with AMC.

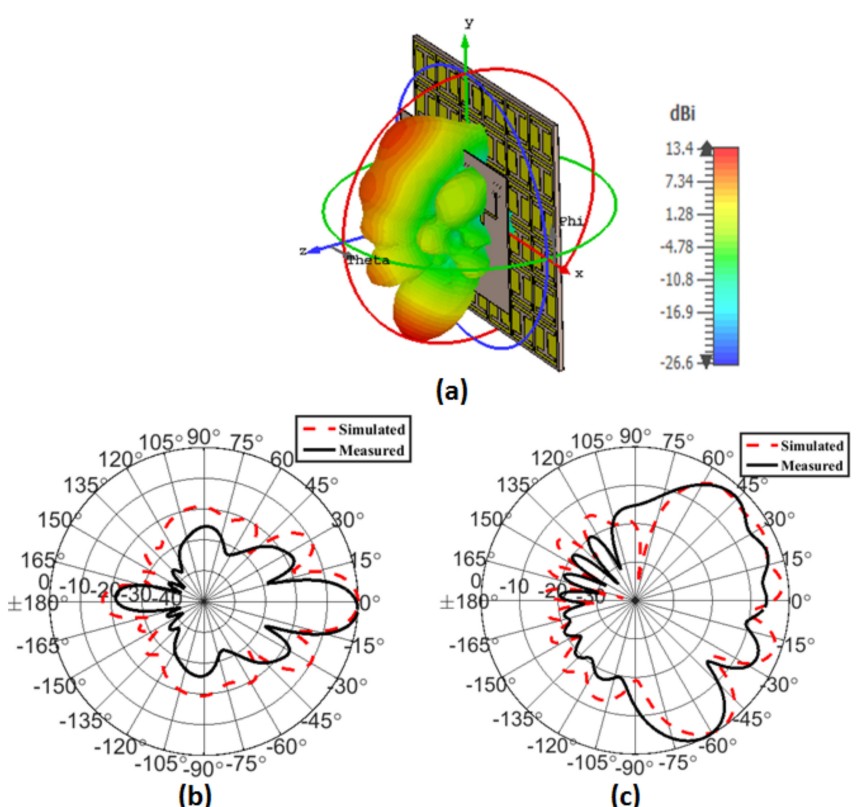

**Figure 17.** The radiation pattern of the 1 × 4 antenna array at 28 GHz; (**a**) 3D, (**b**) normalized pattern in XZ-plane, and (**c**) normalized pattern in YZ-plane.

A comparison between our suggested antenna and other designs is displayed in Table 1. The antenna array achieves higher gain in addition to utilizing only four elements. The matching level, impedance bandwidth, and radiation efficiency are kept within reasonable values. The proposed array provides a compact size, and the gain could be further increased by using 8-elements with a small increment in the total array size.

**Table 1.** Comparison with other reported antennas.

| Ref. | Matching Level (dB) | $f_0$/BW (GHz) | No. of Array Elements | Gain (dBi) | Radiation Efficiency (%) | Substrate $\varepsilon_r$ /Thickness (mm)/ AMC Gap (mm)/ Thickness (mm) | Circuit Size (mm × mm) | Circuit Size (λg × λg) |
|---|---|---|---|---|---|---|---|---|
| [9] | 20 | 28/5 | 2 | 10 | 91 | 3.55/0.203/4.7/0.203 | 47 × 47 | 8.3 × 8.3 |
| [12] | 22 | 28/4 | 8 | 12.7 | 71.45 | 3.5/0.2/-/- | 75 × 150 | 13.1 × 26.2 |
| [13] | 24 | 28/3.66 | 5 | 10 | 92 | 4.3/1.6/-/- | 30 × 19 | 5.8 × 3.7 |
| [14] | 20 | 32.5/11.5 | 8 | 12.5 | NA | 2.2/0.254/-/- | 45 × 45 | 7.2 × 7.2 |
| [20] | 28 | 30/1.06 | 1 | 9.2 | NA | 3/0.254/4.6/0.787 | 17.3 × 17.3 | 2.9 × 2.9 |
| [21] | 20 | 33/7 | 1 | 5.5 | 66.5 | 2.2/0.8/-/- | 30 × 16 | 4.9 × 2.6 |
| [22] | 25 | 30.8/5.7 | 1 | 7.8 | 80 | 3.66/0.5/-/- | 12.8 × 12.8 | 2.5 × 2.5 |
| [27] | 17 | 28/1 | 4 | 12.02 | 82 | 2.2/0.787/-/- | 30 × 35 | 2.8 × 3.26 |
| [28] | 22 | 28/4.1 | 4 | 8.3 | 80 | 3.66/0.76/-/- | 30 × 35 | 2.8 × 3.26 |
| [29] | 36 | 28/2 | 4 | 6.1 | 92 | 2.2/0.787/-/- | 30 × 30 | 2.8 × 2.8 |
| This work (without AMC) | 25 | 28/2.91 | 4 | 10.3 | 94.4 | 3.55/0.203/-/- | 30 × 22.1 | 5.3 × 3.8 |
| This work (with AMC) | 27 | 28/3.1 | 4 | 13.01 | 83.05 | 3.55/0.203/4.7/0.787 | 39.5 × 39.5 | 6.9 × 6.9 |

$f_0$: center frequency; BW: bandwidth; NA: not reported in the reference paper.

## 5. Conclusions

A high gain 1 × 4 linear antenna array for 28 GHz, 5G applications, has been designed and implemented. An AMC layer has been introduced to enhance the radiation characteristics and improve the antenna gain. A chamfered rectangular parasitic element has been adopted in the single element for bandwidth tuning. The measured impedance bandwidth was from 25.36 to 26.03 GHz (0.67 GHz bandwidth) and 26.75 to 28.81 GHz (2.06 GHz bandwidth). The proposed array achieved a 13.1 dBi measured gain and a front-to-back ratio between 15 and 20 dB across the operating band at 28 GHz. Good matching between the simulated and tested results was observed. The attractive specifications of the proposed antenna, such as high gain, and compact size, make it a good candidate for 5G NR communication.

**Author Contributions:** Conceptualization, A.A.I.; methodology, H.Z.; software, O.M.D.; validation, A.A.I., and M.A.A.; formal analysis, A.A.I., and O.M.D.; writing—original draft preparation, A.A.I., and H.Z.; writing—review and editing, S.M.A., N.H.; supervision, M.A.A.; project administration, N.H.; All authors have read and agreed to the published version of the manuscript.

**Funding:** This research received no external funding.

**Institutional Review Board Statement:** Not applicable.

**Informed Consent Statement:** Not applicable.

**Data Availability Statement:** Not applicable.

**Conflicts of Interest:** The authors declare no conflict of interest.

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
