# Peer review of "Slotted Antenna Array with Enhanced Radiation Characteristics for 5G 28 GHz Communications"

_electronics, doi:10.3390/electronics11172664_

Round 1

Reviewer 1 Report

Attached is the comments file

Author Response

Dear Reviewer,

Thank you very much for your constructive comments. It has really helped us in improving our manuscript. The manuscript has been revised based on the light of your comments. The modifications have been highlighted in the revised manuscript.

The point to point response to each comment has been attached as PDF file.

Thank you.

Best Regards,

Authors

Reviewer 2 Report

The paper presents a linear array antenna resonating at 28 GHz of millimeter wave spectrum for 5G communication systems and AMC layer is designed to increase the radiation intensity and improving overall array gain. The presented work achieved the optimal performance in terms of fractional impedance bandwidth, realized gain, efficiency and excellent radiation performance features. The work has a certain worth and add some interest to the readers. However, I have some suggestions to further improve the quality of the manuscript.

1. Overall English writing of the manuscript should proof read from the native speaker.

2. The authors are suggested to highlight the key contributions and novelty of the manuscript in the introdauction.

3. I suggested authors to further add more recent quality research work published in the year 2020- 2022 and incorporate it in the revised manuscript.

4. Figure 2, Figure 5, Figure 8, Figure 15, replace |S11| (dB) to Reflection Coefficient (dB).

5. Figure 8 and Figure 9 should be merged together as you already shown the reflection coefficient and gain plot in Figure 2 and Figure 5. Moreover, I suggest you to maintain the quality of figures throughput the manuscript.

6. I see a lot of discrepancies in the measured results, Why is it so? Please explain the solid reasons of these discrepancies.

7. Figure 10 (a) – (c), I didn’t see the gain axis. How can reader understand the amount of gain of the proposed work? It is suggested to show the gain performance of the antenna array in the polar as well as Cartesian coordinate systems.

8. Again in Figure 17 (a) – (b), I didn’t see the gain axis. Please provide the gain axis so that readers can easily understand the amount of gain.

9. In the comparison table I see some typos particularly in the gain performance with AMC and without AMC. The authors are required to again have a look and modify it accordingly.

Author Response

Dear Reviewer,

Thank you very much for your constructive comments. It has really helped us in improving our manuscript. The manuscript has been revised based on the light of your comments. The modifications have been highlighted in the revised manuscript. The point-to-point response to each comment has been attached as PDF file.

Thank you.

Best Regards,

Authors

Round 2

Reviewer 1 Report

Thanks for considering the comments.